# Distilled Feature Fields Enable Few-Shot Language-Guided Manipulation

**William Shen**[*1], **Ge Yang**[*1,2], **Alan Yu**[1], **Jansen Wong**[1],
**Leslie Pack Kaelbling**[1], **Phillip Isola**[1]

[1]MIT CSAIL, [2]Institute for Artificial Intelligence and Fundamental Interactions

**Abstract:** Self-supervised and language-supervised image models contain rich knowledge of the world that is important for generalization. Many robotic tasks, however, require a detailed understanding of 3D geometry, which is often lacking in 2D image features. This work bridges this 2D-to-3D gap for robotic manipulation by leveraging distilled feature fields to combine accurate 3D geometry with rich semantics from 2D foundation models. We present a few-shot learning method for 6-DOF grasping and placing that harnesses these strong spatial and semantic priors to achieve in-the-wild generalization to unseen objects. Using features distilled from a vision-language model, CLIP, we present a way to designate novel objects for manipulation via free-text natural language, and demonstrate its ability to generalize to unseen expressions and novel categories of objects. Project website: https://f3rm.csail.mit.edu

## 1 Introduction

What form of scene representation would facilitate open-set generalization for robotic manipulation systems? Consider a warehouse robot trying to fulfill an order by picking up an item from cluttered storage bins filled with other objects. The robot is given a product manifest, which contains the text description it needs to identify the correct item. In scenarios like this, geometry plays an equally important role as semantics, as the robot needs to comprehend which parts of the object geometry afford a stable grasp. Undertaking such tasks in unpredictable environments — where items from a diverse set can deviate markedly from the training data, and can be hidden or jumbled amidst clutter — underscores the critical need for robust priors in both spatial and semantic understanding.

In this paper, we study few-shot and language-guided manipulation, where a robot is expected to pick up novel objects given a few grasping demonstrations or text descriptions without having previously seen a similar item. Toward this goal, we build our system around pre-trained image embeddings, which have emerged as a reliable way to learn commonsense priors from internet-scale data [1, 2, 3].

Figure 1 illustrates how our system works. The robot first scans a tabletop scene by taking a sequence of photos using an RGB camera mounted on a selfie stick (Figure 1, left). These photos are used to construct a neural radiance field (NeRF) of the tabletop, which, crucially, is trained to render not just RGB colors but also *image features* from a pre-trained vision foundation model [4, 1]. This produces a scene representation, called a Distilled Feature Field (DFF), that embeds knowledge from 2D feature maps into a 3D volume (Figure 1, middle). The robot then references demonstrations and language instructions to grasp objects specified by a user (Figure 1, right).

Distilled Feature Fields (DFFs) were introduced in computer graphics for tasks such as decomposing and editing images [5, 6]. The main contribution of this work is to study the use of DFFs instead for robotic manipulation. We evaluate the robot's ability to generalize using features sourced from self-supervised vision transformers (DINO ViT, see [4]). These features have been shown to be effective out-of-the-box visual descriptors for dense correspondence [7]. We also source features

---

*Equal contribution. Correspondence to {willshen,geyang}@csail.mit.edu

7th Conference on Robot Learning (CoRL 2023), Atlanta, USA.

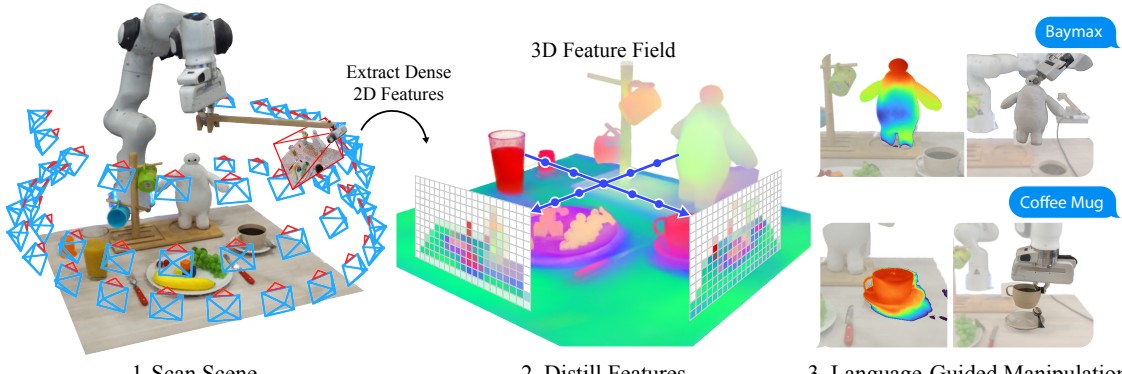

Figure 1: **Distilled Feature Fields Enable Open-Ended Manipulation.** (1) Robot uses a selfie stick to scan RGB images of the scene (camera frustums shown). (2) Extract patch-level dense features for the images from a 2D foundation model, and distill them into a feature field (PCA shown) along with modeling a NeRF. (3) We can query CLIP feature fields with language to generate heatmaps and infer 6-DOF grasps on novel objects given only ten demonstrations.

from a vision-language model, CLIP [1], which is a strong zero-shot learner on various vision and visual question-answering tasks.

One challenge that makes distilled feature fields unwieldy for robotics is the long time it takes to model each scene. To address this, we build upon the latest NeRF techniques, and employ hierarchical hashgrids to significantly reduce the modeling time [8, 9, 10]. When it comes to vision-language features, CLIP is trained to produce image-level features, whereas 3D feature distillation requires dense 2D descriptors. Our solution is to use the MaskCLIP [11] reparameterization trick, which extracts dense patch-level features from CLIP while preserving alignment with the language stream.

We demonstrate that Distilled Feature Fields enable open-ended scene understanding and can be leveraged by robots for 6-DOF object manipulation. We call this approach Feature Fields for Robotic Manipulation (F3RM). We present few-shot learning experiments on grasping and placing tasks, where our robot is able to handle open-set generalization to objects that differ significantly in shape, appearance, materials, and poses. We also present language-guided manipulation experiments where our robot grasps or places objects in response to free-text natural language commands. By taking advantage of the rich visual and language priors within 2D foundation models, our robot generalizes to new categories of objects that were not seen among the four categories used in the demonstrations.

## 2 Problem Formulation

We consider the class of manipulation problems that can be parameterized via a single rigid-body transformation $\mathbf{T} \in SE(3)$, and focus on grasping and placing tasks. We parameterize a 6-DOF grasp or place pose as $\mathbf{T} = (\mathbf{R}, \mathbf{t})$ in the world frame (see Figure 2), where $\mathbf{R}$ is the rotation matrix, and $\mathbf{t}$ is the translation vector. In each scene, the robot is given a set of RGB images $\{\mathbf{I}\}$ with their corresponding camera poses.

**Few-Shot Manipulation.** We aim to build robots that can manipulate objects given only a few demonstrations of a task, such as grasping a mug by its handle. During *learning,* each demonstration $D$ consists of the tuple $\langle\{\mathbf{I}\}, \mathbf{T}^*\rangle$, where $\{\mathbf{I}\}_{i=1}^N$ are $N$ RGB camera views of the scene and $\mathbf{T}^*$ is a pose that accomplishes the desired task. During *testing,* the robot is given multiple images $\{\mathbf{I}'\}$ of a new scene which may contain distractor objects and clutter. The robot's goal is to predict a pose $\mathbf{T}$ that achieves the task. We want to test for open-ended generalization: the new scene contains related but previously unseen objects that differ from the demo objects in shape, size, pose, and material.

**Open-Text Language-Guided Manipulation.** We extend few-shot manipulation to include open-text conditioning via natural language. Given a text description of an object, the robot's objective is to grasp the objects that match this description. The robot has access to a few demonstrations for multiple object categories. During *testing*, the user provides the robot with a text query $L^+$ to specify which object to manipulate and negative texts $L^-$ to reject distractors. In practice, $L^-$ can be sampled automatically. The object category we care about at test time does not necessarily appear in the demonstrations. We explicitly seek open-ended generalization to new object categories, given just a few demonstrations limited to a small class of object categories.

## 3 Feature Fields for Robotic Manipulation (F3RM)

We present *Feature Fields for Robotic Manipulation* (F3RM), our approach for distilling pre-trained representations from vision and vision-language models into 3D feature fields for open-ended robotic manipulation. Doing so involves solving three separate problems. First, how to produce the feature field of a scene automatically at a reasonable speed; second, how to represent and infer 6-DOF grasping and placing poses; and finally, how to incorporate language guidance to enable open-text commands. We include a formal summary of NeRFs [12] in Appendix A.1.

### 3.1 Feature Field Distillation

Distilled Feature Fields (DFFs) [5, 6] extend NeRFs by including an additional output to reconstruct dense 2D features from a vision model $\mathbf{f}_{\text{vis}}$. The feature field $\mathbf{f}$ is a function that maps a 3D position $\mathbf{x}$ to a feature vector $\mathbf{f}(\mathbf{x})$. We assume that $\mathbf{f}$ does not depend on the viewing direction $\mathbf{d}$.[1] Supervision of this feature field $\mathbf{f}$ is provided through the rendered 2D feature maps, where each feature vector is given by the feature rendering integral between the near and far plane ($t_n$ and $t_f$):

$$\mathbf{F}(\mathbf{r}) = \int_{t_n}^{t_f} T(t)\sigma(\mathbf{r}_t)\mathbf{f}(\mathbf{r}_t)\,\mathrm{d}t \quad \text{with} \ \ T(t) = \exp\left(-\int_{t_n}^{t} \sigma(\mathbf{r}_s)\,\mathrm{d}s\right), \quad (1)$$

where $\mathbf{r}$ is the camera ray corresponding to a particular pixel, $t$ is the distance along the ray, $\sigma$ is the density field from NeRF, and $T(t)$ represents the accumulated transmission from the near plane $t_n$ to $t$. Observe this is similar to the volume rendering integral (Eq. 5).

**Feature Distillation.** We begin with a set of $N$ 2D feature maps $\{\mathbf{I}_i^f\}_{i=1}^N$, where $\mathbf{I}^f = \mathbf{f}_{\text{vis}}(\mathbf{I})$ for each RGB image $\mathbf{I}$. We optimize $\mathbf{f}$ by minimizing the quadratic loss $\mathcal{L}_{\text{feat}} = \sum_{\mathbf{r}\in\mathcal{R}} \left\|\hat{\mathbf{F}}(\mathbf{r}) - \mathbf{I}^f(\mathbf{r})\right\|_2^2$, where $\mathbf{I}^f(\mathbf{r})$ is the target feature vector from $\mathbf{I}^f$, and $\hat{\mathbf{F}}(\mathbf{r})$ is estimated by a discrete approximation of the feature rendering integral in Eq. 1.

**Extracting Dense Visual Features from CLIP.** The common practice for extracting dense features from ViTs is to use the key/value embeddings from the last layer before pooling [4]. While this approach has yielded great results for vision-only models on image segmentation and image-to-image correspondence tasks [7, 15], it removes the transformations needed to project the visual features into the shared feature space with the language stream in vision-language models such as CLIP [1, 16]. Re-aligning the dense features typically requires additional training, which negatively affects the model's open-text generalization.

Rather than using CLIP as an image-level feature (Alg. 1, App. A.2), we extract dense features from CLIP using the MaskCLIP reparameterization trick (Alg. 2) [11]. These features retain a sufficient alignment with the language embedding to support zero-shot language guidance in our experiments (see Fig.7). We present pseudocode for this technique in Appendix A.2. A second modification we apply is to interpolate the position encoding (see 4) to accommodate larger images with arbitrary aspect ratios. This is needed because CLIP uses a small, fixed number of input patches from a square crop. These two techniques combined enable us to extract dense, high-resolution patch-level 2D features from RGB images at about 25 frames per second and does not require fine-tuning CLIP.

---

[1]Prior work that uses the association of image pixels in 3D to supervise learning 2D image features makes the same assumption [13, 14].

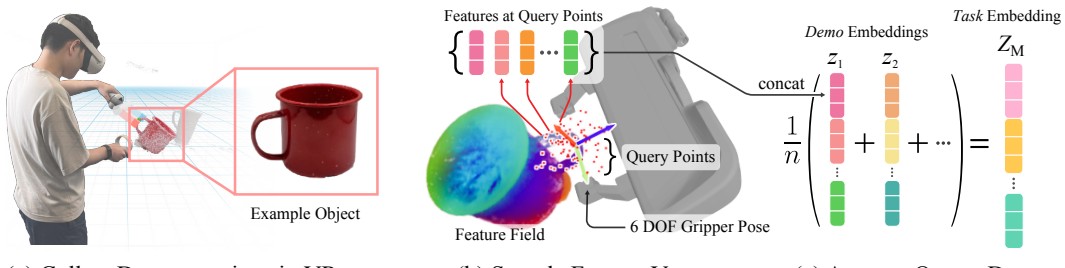

(a) Collect Demonstrations in VR     (b) Sample Feature Vectors     (c) Average Over *n* Demos

Figure 2: **Representing 6-DOF Poses.** (a) Recording the gripper pose $\mathbf{T}^*$ in virtual reality (VR) on an example mug. (b) We approximate the continuous local field via a fixed set of query points in the gripper's canonical frame. (c) We concatenate feature vectors at these query points, then average over $n$ (we use $n = 2$) demonstrations. This gives a task embedding $\mathbf{Z}_M$ for the task $M$.

## 3.2 Representing 6-DOF Poses with Feature Fields

We wish to represent the pose of the gripper in a demonstration by the local 3D feature field in the gripper's coordinate frame. We approximate this local context via a discrete set of query points and the feature vectors measured at each point. We sample a fixed set of $N_q$ query points $\mathcal{X} = \{\mathbf{x} \in \mathbb{R}^3\}_{N_q}$ in the canonical gripper frame for each task $M$ from a 3D Gaussian. We adjust its mean and variance manually to cover parts of the object we intend to target, as well as important context cues (e.g., body of the mug when grasping the handle) and free-space (Fig.2b). For a 6-DOF gripper pose $\mathbf{T}$, we sample the feature field $\mathbf{f}$ at each point in the query point cloud, transformed by $\mathbf{T}$ (Fig.2b).

To account for the occupancy given by the local geometry, we weigh the features by their corresponding alpha values from the density field $\sigma$ of the NeRF model, integrated over the voxel. At a point $\mathbf{x}$ in the world frame, this produces the $\alpha$-*weighted features*

$$\mathbf{f}_\alpha(\mathbf{x}) = \alpha(\mathbf{x}) \cdot \mathbf{f}(\mathbf{x}), \text{ where } \alpha(\mathbf{x}) = 1 - \exp(-\sigma(\mathbf{x}) \cdot \delta) \in (0, 1), \tag{2}$$

and $\delta$ is the distance between adjacent samples. We sample a set of features $\{\mathbf{f}_\alpha(\mathbf{x}) \mid \mathbf{x} \in \mathbf{T}\mathcal{X}\}$ using the transformed query points $\mathbf{T}\mathcal{X}$, and concatenate along the feature-dimension into a vector, $\mathbf{z}_\mathbf{T} \in \mathbb{R}^{N_q \cdot |\mathbf{f}|}$. The query points $\mathcal{X}$ and demo embedding $\mathbf{z}_\mathbf{T}$ thus jointly encode the demo pose $\mathbf{T}$.

We specify each manipulation task $M$ by a set of demonstrations $\{D\}$. We average $\mathbf{z}_\mathbf{T}$ over the demos for the same task to obtain a *task embedding* $\mathbf{Z}_M \in \mathbb{R}^{N_q \cdot |\mathbf{f}|}$ (Fig. 2c). This allows us to reject spurious features and focus on relevant parts of the feature space. This representation scheme is similar to the one used in Neural Descriptor Fields [17]. The main distinction is that NDF is trained from scratch on object point clouds, whereas our feature field is sourced from 2D foundation models that are trained over internet-scale datasets. The capabilities that emerge at this scale hold the potential for open-ended generalization beyond the few examples that appear in the demonstrations.

**Inferring 6-DOF Poses.** Our inference procedure involves a coarse pre-filtering step for the translational DOFs, and an optimization-based fine-tuning step for the rotational DOFs. First, we sample a dense voxel grid over the workspace, where each voxel $\mathbf{v}$ has a grid-size $\delta$. We remove free space by rejecting voxels with alphas $\alpha(\mathbf{v}) < \epsilon_{\text{free}}$. We then remove voxels that are irrelevant to the task, using the cosine similarity between the voxel feature $\mathbf{f}_\alpha(\mathbf{v})$ and the task embedding $\mathbf{Z}_M$. To get the complete 6-DOF poses $\mathcal{T} = \{\mathbf{T}\}$, we uniformly sample $N_r$ rotations for each remaining voxel $\mathbf{v}$.

**Pose Optimization.** We optimize the initial poses with the following cost function

$$\mathcal{J}_{\text{pose}}(\mathbf{T}) = -\cos(\mathbf{z}_\mathbf{T}, \mathbf{Z}_M) \tag{3}$$

using the Adam optimizer [18] to search for poses that have the highest similarity to the task embedding $\mathbf{Z}_M$. After each optimization step, we prune poses that have the highest costs. We also reject poses that are in collision by thresholding the number of overlapping voxels between a voxelized gripper model and the scene geometry. This leaves us with a ranked list of poses that we feed into a motion planner in PyBullet [19, 20]. We execute the highest-ranked grasp or place pose that has a

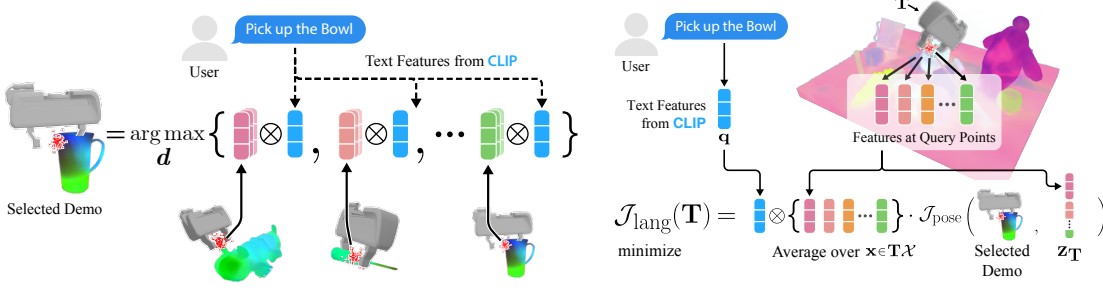

(a) Retrieving Demonstrations           (b) Language-Guided Pose Optimization

Figure 3: **Pipeline for Language-Guided Manipulation.** (a) Encode the language query with CLIP, and compare its similarity to the average query point features over a set of demos. The mug lip demos have the highest similarity to "Pick up the Bowl". (b) Generate and optimize grasp proposals using the CLIP feature field by minimizing $\mathcal{J}_{\text{lang}}$. We use the selected demo from (a) in $\mathcal{J}_{\text{pose}}$, and compute the language-guidance weight with the text features and average query point features.

valid motion plan. Observe that our scheme operates over the entire feature field, and does not rely on assumptions about objectness such as segmentation masks or object poses.

### 3.3    Open-Text Language-Guided Manipulation

Natural language offers a way to extend robotic manipulation to an open-set of objects, serving as an attractive alternative when photos of the target object are inaccurate or unavailable. In our language-guided few-shot manipulation pipeline, the learning procedure and the representation for the demonstrations remain consistent with Section 3.2. At test time, the robot receives open-text language queries from the user that specify the object of interest to manipulate. Our language-guided pose inference procedure comprises three steps (see Fig.3): (i) retrieving relevant demonstrations, (ii) initializing coarse grasps, and (iii) language-guided grasp pose optimization.

**Retrieving Relevant Demonstrations.** We select the two demonstrations whose average feature $\mathbf{F}_d$ (averaged among the query points of each demo pose $\mathbf{T}^*$) is closest to the text embedding $\mathbf{q} = \text{emb}_{\text{CLIP}}(L^+)$ (Fig.3a). We found that using the positive query text ($L^+$) alone is sufficient. This means finding the demonstration that maximizes the cosine similarity $\cos(\mathbf{q}, \mathbf{F}_d)$. Note that the objects used in the demonstrations do not have to come from the same category as the target object. For instance, asking the robot to pick up the "measuring beaker" or "bowl" leads to the robot choosing the demonstration of picking up a mug by its lip (Fig.4).

**Initializing Grasp Proposals.** We speed up grasp pose inference by first running a coarse proposal step where we filter out regions in the feature field that are irrelevant to the text query. We start by sampling a dense voxel grid among the occupied regions by masking out free space (see Sec.3.2). Afterward, we prune down the number of voxels by keeping those more similar to the positive query $L^+$ than any one of the negative queries $L^-$. Formally, let $\mathbf{q}_i^- = \text{emb}_{\text{CLIP}}(L_i^-) \mid i \in \{1, \ldots, n\}$ be the text embeddings of the negative queries. We compute the softmax over the pair-wise cosine similarity between the voxel's feature $\mathbf{f}_\alpha(\mathbf{v})$ and the ensemble $[\mathbf{q}, \mathbf{q}_1^-, \mathbf{q}_2^-, \ldots, \mathbf{q}_n^-]$, and identify the closest negative query $\mathbf{q}^-$. We remove voxels that are closer to $\mathbf{q}^-$ than the positive query $\mathbf{q}$. The cosine similarity between the voxel embedding and $[\mathbf{q}, \mathbf{q}^-]$ pair forms a binomial distribution that allows us to reject voxels that have $< 50\%$ probability of being associated with the negative query. Finally, to get the set of initial poses $\mathcal{T} = \{\mathbf{T}\}$, we sample $N_r$ rotations for each remaining voxel.

**Language-Guided Grasp Pose Optimization.** To incorporate language guidance, we first compute $\mathcal{J}_{\text{pose}}$ from Eq.3 using the two demonstrations retrieved in the first step. We then assign a lower cost to regions that are more similar to the language query $\mathbf{q}$ by computing a language-guidance weight $C_{\mathbf{q}} = \text{mean}_{\mathbf{x} \in \mathbf{T}\mathcal{X}}[\mathbf{q} \otimes \mathbf{f}_\alpha(\mathbf{x})]$, and multiply it with $\mathcal{J}_{\text{pose}}$ (Fig.3b)

$$\mathcal{J}_{\text{lang}}(\mathbf{T}) = \underset{\mathbf{x} \in \mathbf{T}\mathcal{X}}{\text{mean}} \Big[ \mathbf{q} \otimes \mathbf{f}_\alpha(\mathbf{x}) \Big] \cdot \mathcal{J}_{\text{pose}}(\mathbf{T}). \tag{4}$$

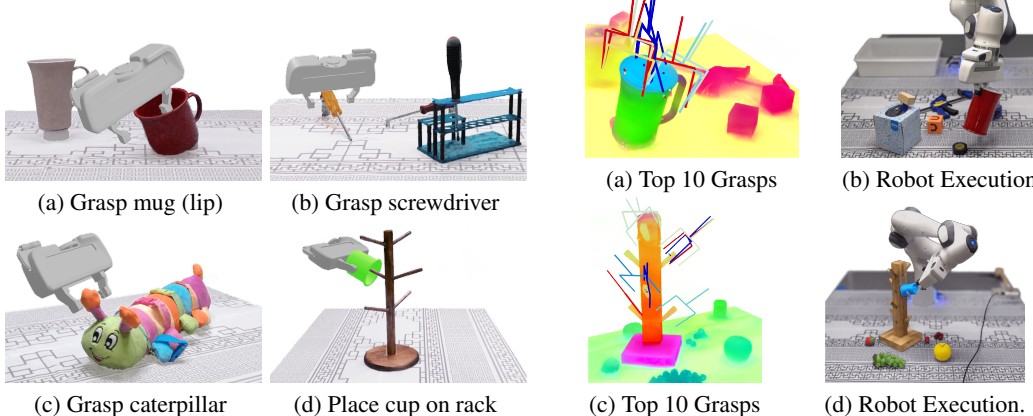

| (a) Grasp mug (lip) | (b) Grasp screwdriver |
| --- | --- |
| (c) Grasp caterpillar | (d) Place cup on rack |

| (a) Top 10 Grasps | (b) Robot Execution |
| --- | --- |
| (c) Top 10 Grasps | (d) Robot Execution. |

Figure 4: **Five Grasping and Place Tasks.** (a) grasping a mug by its lip or handle (Fig.2); (b) a screwdriver by the handle; (c) the caterpillar by its ears; and (d) placing a cup onto a drying rack. Gripper poses indicate one of two demonstrations.

Figure 5: **Generalizing to Novel Objects.** (Top Row) Mug is much bigger than the ones used for demonstration. (Bottom Row) This rack has shorter pegs with a square cross-section. Demo rack is cylindrical (cf. Fig.4d).

The first term, $C_{\mathbf{q}}$, is the normalized cosine similarity between the text embedding $\mathbf{q}$ and the average $\alpha$-weighted query point feature for a pose $\mathbf{T}$. We iteratively update the pose $\mathbf{T}$ via gradient descent while pruning using the procedure from Section 3.2 till convergence.

## 4 Results

### 4.1 Learning to Grasp from Demonstrations

We consider five 6-DOF grasping and placing tasks and provide two demonstrations per task (Fig.4). To label the demonstrations, we load a NeRF-reconstructed point cloud into virtual reality, and use a hand controller to move the gripper to the desired pose (Fig.2a). We compare the performance of three types of distilled features: (1) DINO ViT, (2) CLIP ViT, and (3) CLIP ResNet. We consider three baselines, including (1) using density $\sigma$ from the NeRF, (2) the intermediate NeRF features, and (3) the RGB color value as features, and compare against MIRA [21], a recent work which uses NeRFs to render orthographic viewpoints for pixel-wise affordance predictions. For each task, we evaluate in ten scenes that contain novel objects in arbitrary poses and distractor objects. The novel objects belong to the same or related object category as the demo objects, but differ in shape, size, material and appearance. We reset the scenes to about the same configuration for each compared method. We include the full details on the experimental setup in Appendix A.4.

We present the success rates in Table 1 and examples of robot executions in Figure 5. While the baselines using density, RGB color values, or intermediate features from NeRF achieve respectable performance, they struggle to identify the semantic category of the objects we care about, especially in complex scenes with more distractors. We find that DINO and CLIP feature fields exhibit impressive generalization capabilities and have complementary advantages. The DINO ViT has a good part-level understanding of object geometry with 7/19 failure cases caused by inaccuracies in the grasp rotations and occasionally, the translations. In comparison, 21/27 failures for CLIP ViT and ResNet combined may be attributed to this issue. We find that CLIP favors semantic and categorical information over geometric features, which are essential for grasping and placing objects. DINO, on the other hand, struggles with distinguishing target objects from distractor objects that contain similar visual appearance to the objects used in the demonstrations. CLIP struggles less in this regard. The fusion between semantic features and detailed 3D geometry offers a way to model multiple objects piled tightly together: in Figure 6b, for instance, a caterpillar toy is buried under other toys. Figure 6c shows our robot grasping the caterpillar, and dragging it from the bottom of the pile.

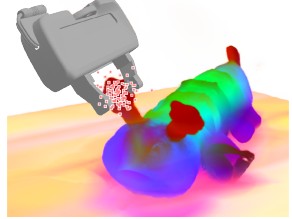
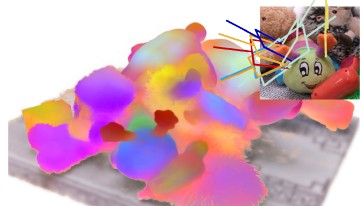
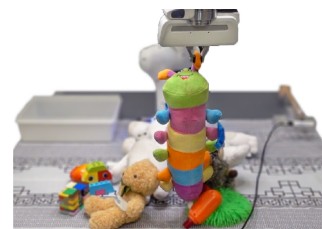

(a) Demonstration (1 of 2)  (b) Feature Field of Cluttered Scene  (c) Robot Execution

Figure 6: **Grasping in a Cluttered Scene.** (a) Demonstration for grasping the caterpillar in its DINO feature field (color is PCA, red dots show query points). (b) A cluttered scene with several toys on top of each other. Inset shows the top 10 inferred grasps. Observe the caterpillar's ears share the same features with the demo. (c) Robot successfully grasps the caterpillar.

|  | Mug lip | Mug handle | Caterpillar ear | Screwdriver handle | Cup on rack | Total |
|---|---|---|---|---|---|---|
| **MIRA [21]** | 1/10 | 2/10 | 6/10 | 3/10 | 3/10 | 15/50 |
| **Density** | 5/10 | 5/10 | **10/10** | 2/10 | 5/10 | 27/50 |
| **Intermediate** | 2/10 | 2/10 | 1/10 | 3/10 | 1/10 | 9/50 |
| **RGB** | 4/10 | 3/10 | 9/10 | 1/10 | 4/10 | 21/50 |
| **DINO ViT** | 5/10 | 4/10 | 8/10 | 6/10 | **8/10** | 31/50 |
| **CLIP ViT** | 7/10 | **7/10** | 8/10 | 6/10 | 6/10 | 34/50 |
| **CLIP ResNet** | **9/10** | 6/10 | 9/10 | **8/10** | 7/10 | **39/50** |

Table 1: **Success rates on grasping and placing tasks.** We compare the success rates over ten evaluation scenes given two demonstrations for each task. We consider a run successful if the robot grasps or places the correct corresponding object part for the task.

| Color | 7/10 |
|---|---|
| Material | 7/10 |
| Relational | 4/10 |
| General | 4/10 |
| OOD | 9/10 |
| **Total** | 31/50 |

Table 2: **Success rates of Language-Guided Manipulation.** Language query success rates across semantic categories.

## 4.2 Language-Guided Object Manipulation

We set up 13 table-top scenes to study the feasibility of using open-text language and CLIP feature fields for designating objects to manipulate. We reuse the ten demonstrations from the previous section (Sec. 4.1), which span four object categories (see Fig.4). We include three types of objects in our test scenes: (1) novel objects from the same categories as the demonstrations, (2) out-of-distribution (OOD) objects from new categories that share similar geometry as the demonstrated items (e.g., bowls, measuring beakers, utensils), and (3) distractor items that we desire the system ignore. **Success metric**: we consider a language query successful if the robot stably grasps the target object and places it in a plastic bin at a known location. We include more details in Appendix A.7.

We break down the success rates by category in Table 2, and show the robot's execution sequence for an example scene in Figure 7 (video). This scene contained eleven objects, four were sourced from the YCB object dataset (the apple, black marker, mango, and a can of SPAM), the rest collected from the lab and bought online. We present five successful grasps (Figure 7). The robot failed to grasp the stainless steel jug by its handle due to a small error in the grasp rotation. This is a typical failure case — six out of 19 failures stem from these poor grasp predictions with rotational or translational errors. The remaining 13/19 failed grasps are due to CLIP features behaving like a bag-of-words and struggling to capture relationships, attributes, and ordinal information within sentences [22]. For instance, in queries such as "black screwdriver" and "mug on a can of spam," CLIP paid more attention to other black objects and the can of spam, respectively. We found that retrieving demos via text occasionally (particularly in the rack scene) benefits from prompt engineering.

In total, our robot succeeds in 31 out of 50 language queries, including both fairly general queries (e.g., mug, drying rack) and ones that specify properties, such as color, material, and spatial relations (e.g., screwdriver on the block). Notably, our robot generalizes to out-of-distribution object categories including bowls, rolls of tape, whiteboard markers and utensils using demonstrations only on mugs and screwdrivers. Although this success rate is far from practical for industrial use, our overall strategy of using 2D visual priors for 3D scene understanding can leverage the rapid advancements in VLMs, which hold significant potential for improving performance.

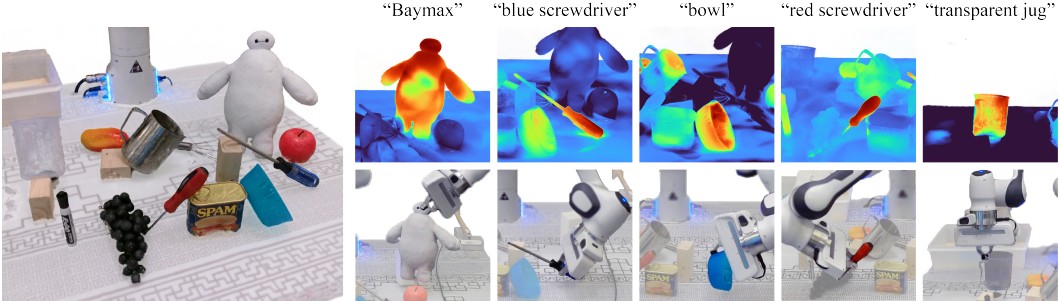

Figure 7: **Language-Guided Manipulation Execution.** (Top Row) Heatmaps given the language queries. (Bottom Row) Robot executing grasps sequentially without rescanning. CLIP can behave like a bag-of-words, as shown by the bleed to the blue bowl for "blue screwdriver."

## 5 Related Work

**Open-Ended Generalization via Language.** A number of prior work use natural language for task-specification and long-horizon planning in applications such as tabletop and mobile manipulation [23, 24, 25, 26] and navigation [27, 28]. A recent line of work seeks to replicate the success of vision and language models in robotics, by jointly pre-training large foundation models on behavior data [29, 30]. The goal of our work is different. F3RM seeks to incorporate geometric information with *any* pre-trained features by lifting imagery and language prior knowledge into 3D.

**Dense 2D Visual Descriptors.** [31, 32] use dynamic 3D reconstruction from RGB-D videos to provide association labels between pixels from different video frames. Dense Object Nets eliminate the need for dynamic reconstruction, using multi-view RGB-D images of static scenes in the context of robotics [13, 33]. NeRF-Supervision extends descriptor learning to thin structures and reflective materials via NeRFs for 3D reconstruction [14]. In contrast, recent work indicates that excellent dense correspondence can emerge at a larger scale without the need for explicit supervision [4, 34].

**Geometric Aware Representations for Robotics.** Geometric understanding is an essential part of mapping [35, 36, 37], grasping [38, 39, 17, 40], and legged locomotion [41]. These work either require direct supervision from 3D data such as point clouds, or try to learn representations from posed 2D images or videos [13, 14]. [21, 42, 43] leverage neural scene representations to take advantage of their ability to handle reflective or transparent objects and fine geometry. Our work incorporates pre-trained vision foundation models to augment geometry with semantics.

**3D Feature Fields.** A number of recent work integrate 2D foundation models with 3D neural fields in contexts other than robotic manipulation [44, 45, 46, 47, 48]. See Appendix A.8 for a comprehensive overview. Our work shares many similarities with LERF [49]. However, unlike the multi-scale image-level features used in LERF, we extract dense patch-level features from CLIP. Additionally, we take a step further by exploring the utilization of feature fields for robotic manipulation.

## 6 Conclusion

We have illustrated a way to combine 2D visual priors with 3D geometry to achieve open-ended scene understanding for few-shot and language-guided robot manipulation. Without fine-tuning, Distilled Feature Fields enable out-of-the-box generalization over variations in object categories, material, and poses. When the features are sourced from vision-language models, distilled feature fields offer language-guidance at various levels of semantic granularity.

**Limitations.** Our system takes 1m 40s to collect 50 images of the scene, and 90s to model the NeRF and feature field. This highlights the need to develop generalizable NeRFs that can recover geometry quickly with just a few views [9, 43], opening the possibility for closed-loop dynamic manipulation. More generally, novel view synthesis is a generative process not too different from image generation with GANs [50] and diffusion models [51]. These alternatives, to which our philosophy equally applies, hold promise for solving general-purpose visual and geometric understanding.

## Acknowledgement

We gratefully acknowledge support from Amazon.com Service LLC, Award #2D-06310236; from the National Science Foundation under Cooperative Agreement PHY-2019786 (The NSF AI Institute for Artificial Intelligence and Fundamental Interactions, http://iaifi.org/); from NSF grant 2214177; from AFOSR grant FA9550-22-1-0249; from ONR MURI grant N00014-22-1-2740; from ARO grant W911NF-23-1-0034; from the MIT-IBM Watson Lab; and from the MIT Quest for Intelligence. The authors also thank Tomás Lozano-Pérez, Lin Yen-Chen and Anthony Simeonov for their advice; Boyuan Chen for initial discussions; Rachel Holladay for her extensive help with setting up the robot; and Tom Silver for providing feedback on an earlier draft.

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

# Appendix

## A.1 Neural Radiance Fields (NeRFs)

Neural radiance fields [12] model a scene as a 6D, vector-valued continuous function that maps from a position $\mathbf{x} = (x, y, z)$ and a normalized viewing direction $\mathbf{d} = (d_x, d_y, d_z)$, to the differential density $\sigma$ and emitted color $(r, g, b)$. In practice, this is achieved via two neural networks which partially share parameters: 1) the density network $\sigma(\mathbf{x})$ which depends only on the position $\mathbf{x}$; and 2) the color network $\mathbf{c}(\mathbf{x}, \mathbf{d})$ which depends on both the position $\mathbf{x}$ and viewing direction $\mathbf{d}$.

**Novel-View Synthesis.** NeRF synthesizes an image by casting a ray $\mathbf{r}$ from the camera origin $\mathbf{o}$ through the center of each pixel. Points along the ray are parameterized as $\mathbf{r}_t = \mathbf{o} + t\mathbf{d}$, where $t$ is the distance of the point to the camera origin $\mathbf{o}$. The color $\mathbf{C}(\mathbf{r})$ of the ray $\mathbf{r}$ between the near and far scene bounds $t_n$ and $t_f$ is given by the volume rendering integral [52]

$$\mathbf{C}(\mathbf{r}) = \int_{t_n}^{t_f} T(t)\sigma(\mathbf{r}_t)\mathbf{c}(\mathbf{r}_t, \mathbf{d}) \, \mathrm{d}t, \quad T(t) = \exp\left(-\int_{t_n}^{t} \sigma(\mathbf{r}_s) \, \mathrm{d}s\right), \tag{5}$$

where $T(t)$ is the accumulated transmittance along the ray from $\mathbf{r}_{t_n}$ to $\mathbf{r}_t$.

**Modeling a Scene with NeRFs.** For a scene, we are given a dataset of $N$ RGB images $\{\mathbf{I}\}_{i=1}^{N}$ with camera poses $\{\mathbf{T}\}_{i=1}^{N}$. At each iteration, we sample a batch of rays $\mathcal{R} \sim \{\mathbf{T}\}_{i=1}^{N}$ and optimize $\sigma$ and $\mathbf{c}$ by minimizing the photometric loss $\mathcal{L}_{\text{rgb}} = \sum_{\mathbf{r} \in \mathcal{R}} \| \hat{\mathbf{C}}(\mathbf{r}) - \mathbf{I}(\mathbf{r}) \|_2^2$, where $\mathbf{I}(\mathbf{r})$ is the RGB value of the pixel corresponding to ray $\mathbf{r} \in \mathcal{R}$, and $\hat{\mathbf{C}}(\mathbf{r})$ is the color estimated by the model using a discrete approximation of Equation 5 [12, 53].

## A.2 Dense 2D Feature Extraction via MaskCLIP

We provide pseudo code for the MaskCLIP method [11] for extracting dense, patch-level features from the CLIP model [1] below. Algorithm 1 is the computation graph of the last layer of vanilla CLIP. Algorithm 2 is MaskCLIP's modified graph. Note that the two linear transformations via $W_{\text{v}}$ and $W_{\text{out}}$ can be fused into a single convolution operation. We provide our feature extraction code in our GitHub repository (https://github.com/f3rm/f3rm).

**Algorithm 1** Image Feature (Original)

```
1 def forward(x):
2   q, k, v = W_qkv @ self.ln_1(x)
3   v = (q[:1] * k).softmax(dim=-1) * v
4   x = x + W_out @ v
5   x = x + self.mlp(self.ln_2(x))
6   return x[:1]    # the CLS token
```

**Algorithm 2** Dense Features (MaskCLIP [11])

```
1 def forward(x):
2   v = W_v @ self.ln_1(x)
3   z = W_out @ v
4   return z[1:] # all but the CLS token
```

## A.3 Feature Fields

**Implementation Details.** Memory for caching the 2D feature map is a significant system bottleneck that does not appear with RGB reconstruction because high-dimensional features, up-scaled to the RGB image resolution, can grow to more than 40 GB for a standard NeRF dataset. We solve this issue by reconstructing patch-level feature maps without up-scaling them to pixel resolution. We speed up our feature distillation by building off newer NeRF implementations using hierarchical hash grids [8] based on Nerfacto [10].

**Feature Field Quality.** F3RM benefits from neural feature fields' ability to reconstruct detailed 3D geometry. We offer such an example in Figure A8. Notice the difference in resolution, between the source 2D feature map (middle), and the final feature field.

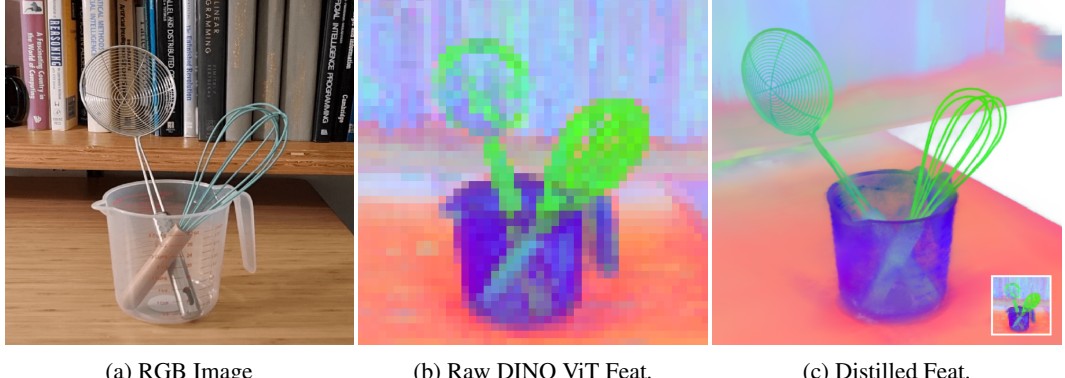

|(a) RGB Image|(b) Raw DINO ViT Feat.|(c) Distilled Feat.|

Figure A8: **Level of Detail.** (a) Mesh strainer and whisk. (b) Raw feature map from DINO ViT, very low in resolution. Colors correspond to PCA of the features. (c) 3D feature fields recover a higher level of detail than the source 2D feature maps. Inset corresponds to (b) in its original size for comparison.

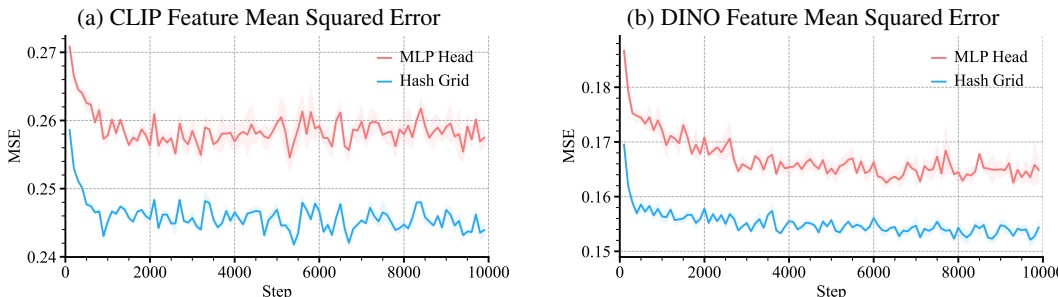

Figure A9: **Feature Error During Feature Distillation.** The mean squared error on a held-out set of feature maps for (a) CLIP and (b) DINO using the MLP head and hash grid architectures described in Section A.3.1. The hash grid architecture consistently achieves a lower error.

### A.3.1  Ablation on Feature Field Architecture

We implement our feature field as a hierarchical hash grid [8] that takes a 3D position **x** as input, and outputs the feature vector. We compare this against a MLP head that takes the intermediate features output by NeRF as input, which is similar to the architectures in [5, 6]. We first train a NeRF on images collected by the robot of a tabletop scene, then distill features for 10000 steps over 3 seeds.

Figure A9 shows that the hash grid architecture achieves a lower mean squared error (MSE), because it is able to capture higher-frequency signals and finer details. While the difference in the MSE seems marginal between these two architectures, the hash grid-based architecture qualitatively results in significantly more well-defined semantic boundaries between objects as shown in Figure A10.

## A.4  Experimental Setup

We provide details about our experimental setup used across our experiments for learning to grasp from demonstrations and language-guided object manipulation.

**Physical Setup.**  We collect RGB images with a RealSense D415 camera (the depth sensor is not used) mounted on a selfie stick. The selfie stick is used to increase the coverage of the workspace, as a wrist-mounted camera can only capture a small area of the workspace due to kinematic limitations. We program a Franka Panda arm to pick up the selfie stick from a magnetic mount, scan $50 \times 1280 \times 720$ RGB images of the scene following a fixed trajectory of three helical passes at different heights, and place the selfie stick back on the mount.

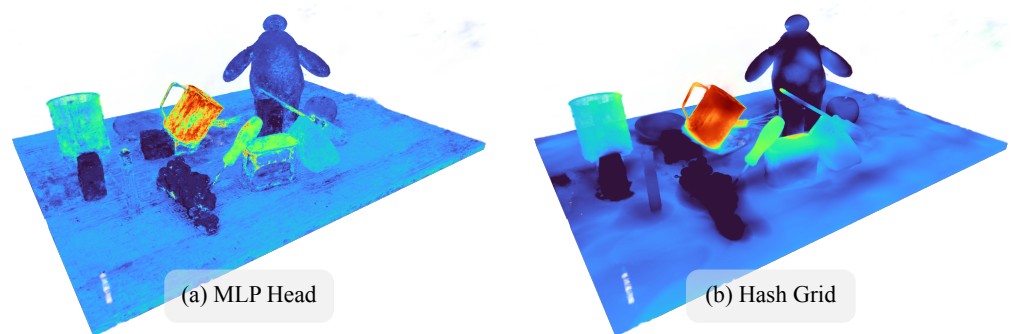

(a) MLP Head       (b) Hash Grid

Figure A10: **Comparing Feature Field Architectures.** We show the similarity heatmaps for the language query "metal mug" on the scene shown in Fig.7. (a) When using the MLP head, regions unrelated to the metal mug exhibit high similarity, as shown by the red bleed onto objects including the red screwdriver and tip of the whiteboard marker. (b) In contrast, the hash grid architecture results in significantly less bleed and more well-defined semantic boundaries between objects.

To calibrate the camera poses, we run COLMAP [54, 55] on a scan of a dedicated calibration scene with calibration markers placed by the robot at known poses. We use these objects to solve for the transformation from the COLMAP coordinate system to the world coordinate system. These camera poses are reused on subsequent scans. Given that the true camera poses vary due to small differences in how the robot grasps the selfie-stick, we optimize them as part of NeRF modeling to minimize any errors and improve reconstruction quality [10, 56, 57].

**Labeling Demonstrations.** We label demonstrations in virtual reality (VR) using a web-based 3D viewer based on Three.js that we developed which supports the real-time rendering of NeRFs, point clouds, and meshes. Given a NeRF of the demonstration scene, we sample a point cloud and export it into the viewer. We open the viewer in a Meta Quest 2 headset to visualize the scene, and move a gripper to the desired 6-DOF pose using the hand controllers (see Fig.2a).

**NeRF and Feature Field Modeling.** We downscale the images to $640 \times 480$ to speed up modeling of the RGB NeRF, and use the original $1280 \times 720$ images as input to the vision model for dense feature extraction. We optimize the NeRF and feature field sequentially for 2000 steps each, which takes at most 90s (average is 80s) on a NVIDIA RTX 3090, including the time to load the vision model into memory and extract features.

| Feature Type | Resolution |
|---|---|
| DINO ViT | $98 \times 55$ |
| CLIP ViT | $42 \times 24$ |
| CLIP ResNet | $24 \times 14$ |

Table 3: **Feature Map Resolutions.** Resolutions of the features output by the vision models given a $1280 \times 720$ RGB image.

In our experiments, we distill the features at their original feature map resolution which is significantly smaller than the RGB images (see Table 3). We achieve this by transforming the camera intrinsics to match the feature map resolutions, and sampling rays based on this updated camera model. The specific models we used were `dino_vits8` for DINO ViT, `ViT-L/14@336px` for CLIP ViT, and `RN50x64` for CLIP ResNet.

## A.5 Ablation on Number of Training Views

Although our robot scans 50 images per scene in our experiments, we demonstrate that it is possible to use a significantly smaller number of views for NeRF and feature field modeling without a significant loss in quality. To investigate this, we ablate the number of training images by evenly subsampling from the 50 scanned images and modeling a NeRF and feature field.

Figure A11 qualitatively compares the RGB, depth, and segmentation heatmaps. We observe an increase in floaters as we reduce the number of training images, with approximately 20 images being the lower bound before a drastic decline in quality.

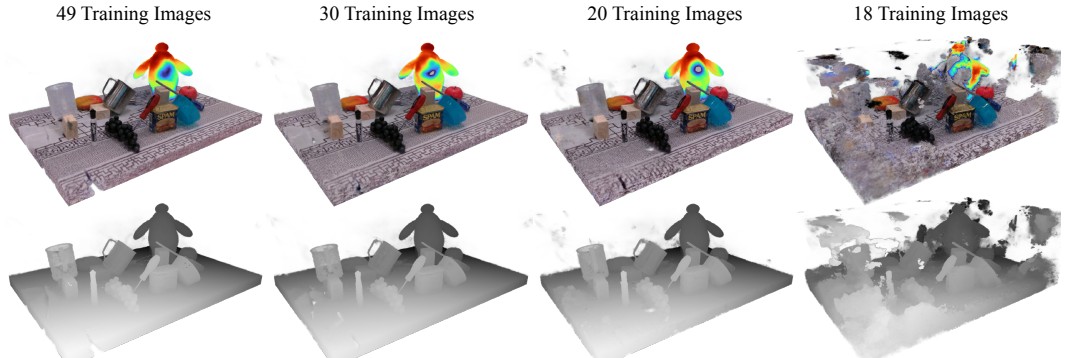

Figure A11: **Ablating Number of Training Views.** We qualitatively compare feature fields trained on different numbers of views. (Top Row) Segmentation heatmap for "Baymax" from a CLIP feature field overlaid on the RGB image from NeRF. (Bottom Row) Depth map rendered from NeRF.

## A.6   Learning to Grasp from Demonstrations

**Sampling Query Points.**   We use $N_q = 100$ query points across all the tasks shown in Fig.4. As other works have observed [17, 58], the downstream performance can vary significantly across different samples of the query points. To address this issue, we sample five sets of query points over different seeds for each task, and run the grasp optimization procedure across a set of test scenes used for method development. We select the query points that achieved the highest success rate on the test scenes. The covariance of the Gaussian is manually tuned to fit the task.

**Grasp Pose Optimization.**   We first discuss how we initialize the grasp poses. We consider a tabletop workspace of size $0.7 \times 0.8 \times 0.35$ meters, and sample a dense voxel grid over the workspace with voxels of size $\delta = 0.0075$m (we use $0.005$m for the cup on racks experiment), where each voxel $\mathbf{v} = (x, y, z)$ represents the translation for a grasp pose.

Next, we compute the alpha value $\alpha(\mathbf{v})$ for each voxel using the NeRF density network $\sigma$, and filter out voxels with $\alpha(\mathbf{v}) < \epsilon_{\text{free}} = 0.1$. This removes approximately $98\%$ of voxels by ignoring free space. The cosine similarity of the voxel features $\mathbf{f}_\alpha(\mathbf{v})$ is thresholded with the task embedding $\mathbf{Z}_M$ to further filter out voxels. This threshold is adjusted depending on the task and type of feature distilled, and typically cuts down $80\%$ of the remaining voxels. Finally, we uniformly sample $N_r = 8$ rotations for each voxel to get the initial grasp proposals $\mathcal{T}$.

We minimize Equation 3 to find the grasp pose that best matches the demonstrations using the Adam optimizer [18] for 50 steps with a learning rate of 5e-3. This entire procedure takes 15s on average, but could easily be sped up.

**Grasp Execution.**   We reject grasp poses which cause collisions by checking the overlap between a voxelized mesh of the Panda gripper and NeRF geometry by querying the density field $\sigma$. We input the ranked list of grasp poses into an inverse kinematics solver and BiRRT motion planner in PyBullet [19, 20], and execute the highest-ranked grasp with a feasible motion plan.

### A.6.1   Baselines

We provide implementation details of the four baselines used in our few-shot imitation learning experiments. The first three baselines use NeRF-based outputs as features for the query point-based pose optimization:

1. Density: we use the alpha $\alpha \in (0, 1)$ values for NeRF density to ensure the values are scaled consistently through different scenes, as the density values output by the density field $\sigma$ are unbounded.

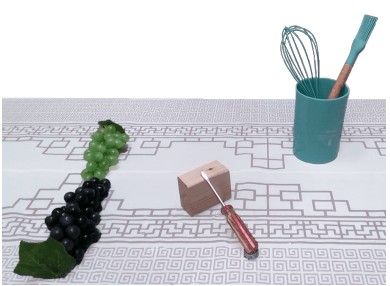
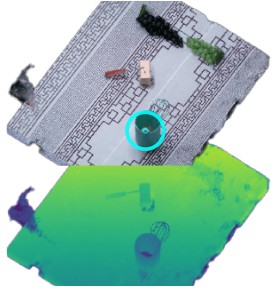
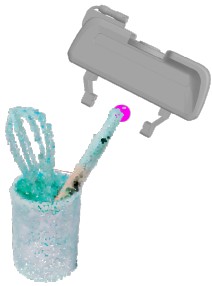

(a) Test Scene with a Screwdriver     (b) Affordance Prediction     (c) Predicted Grasp

Figure A12: **MIRA Failure Case.** (a) Test scene with a screwdriver and other distractors for the screwdriver task (Fig.4b). (b) The orthographic render of the view selected by MIRA, we show the RGB (top) and depth (bottom) renders. The pixel circled in cyan indicates the action with the highest pixel-wise affordance across all views. (c) The predicted 6-DOF grasp incorrectly targets the silicone brush, as it shares resemblance to a screwdriver from a top-down perspective.

2. Intermediate Features: we use the features output by the intermediate density embedding MLP in Nerfacto [10], which have a dimensionality of 15.

3. RGB: we use $[r, g, b, \alpha]$ as the feature for this baseline. $\alpha$ is used to ensure that this baseline pays attention to both the color and geometry, as we found that using RGB only with the alpha-weighted feature field $\mathbf{f}_\alpha$ (Eq.2) collapsed RGB values to $(0, 0, 0)$ for free space, which corresponds to the color black.

**MIRA Baseline.** The fourth baseline we consider is Mental Imagery for Robotic Affordances (MIRA) [21], a NeRF-based framework for 6-DOF pick-and-place from demonstrations that renders orthographic views for pixel-wise affordance prediction. Orthographic rendering ensures that an object has the same size regardless of its distance to the camera, and is used to complement the translation equivariance of the Fully Convolutional Network (FCN) for predicting affordances.

MIRA formulates each pixel in a rendered orthographic view as a 6-DOF action $\mathbf{T} = (\mathbf{R}, \mathbf{t})$, with the orientation of the view defining the rotation $\mathbf{R}$ and the estimated depth from NeRF defining the translation $\mathbf{t}$. The FCN is trained to predict the pixels in the rendered views corresponding to the demonstrated 6-DOF actions, and reject pixels sampled from a set of negative views. During inference, MIRA renders several orthographic views of the scene and selects the pixel that has the maximum affordance across all views.

In our experiments, we train a separate FCN for 20000 steps for each task in Fig.4 specified by two demonstrations, and sample negative pixels from datasets consisting solely of distractor objects. We use data augmentation following Yen-Chen et al. [21]'s provided implementation and apply random SE(2) transforms to the training views. Given a test scene, we scan 50 RGB images as described in Appendix A.4, render 360 orthographic viewpoints randomly sampled over an upper hemisphere looking towards the center of the workspace, and infer a 6-DOF action.

MIRA was designed for suction cup grippers and does not predict end-effector rotations. We attempted to learn this rotation, but found that the policy failed to generalize. To address this issue and give MIRA the best chance of success, we manually select the best end-effector rotation to achieve the task. We additionally find that MIRA often selects floater artifacts from NeRF, and manually filter these predictions out along with other unreasonable grasps (e.g., grasping the table itself).

Given that MIRA is trained from scratch given just two demonstrations, we find that it struggles to generalize and is easily confused by floaters and distractors despite data augmentations and negative samples. MIRA additionally reasons over 2.5D by using the rendered RGB and depth from NeRF as inputs to the FCN, while our query point-based formulation reasons explicitly over 3D. Because of this, we observe that MIRA can fail when there are occlusions or distractor objects that look

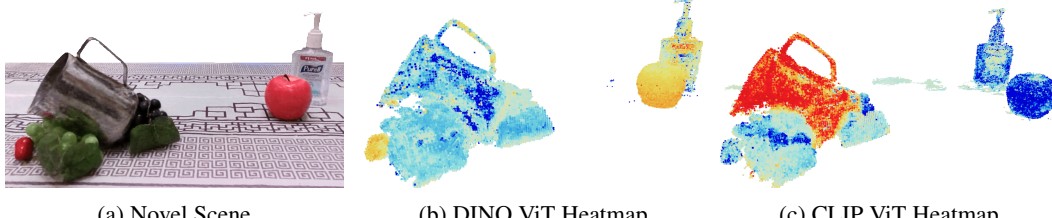

|     (a) Novel Scene     |     (b) DINO ViT Heatmap     |     (c) CLIP ViT Heatmap     |

Figure A13: **Comparing DINO and CLIP feature fields.** We depict the cosine similarity for the task of grasping a mug by the handle. Two demos are provided on a red and a white mug (cf. Fig.3b). (b) DINO overfits to the red color of the apple, while (c) CLIP captures higher-level semantics, and identifies the metal mug.

like the demonstration objects from certain viewpoints. For example, one of the demonstrations for the screwdriver task was a top-down grasp on a screwdriver standing vertically in a rack (Fig.4b). Figure A12 depicts a test scene for the screwdriver grasping task where MIRA incorrectly selects the silicone brush as it looks similar to a screwdriver from a top-down 2.5D view. DINO and CLIP ResNet feature fields successfully grasp the screwdriver in this scene, highlighting the benefits of using pretrained features and reasoning explicitly over 3D geometry.

### A.6.2   DINO Failure Cases

Our experiments show that DINO struggles with distractor objects which have high feature similarity to the demonstrations, despite not representing the objects and their parts we care about (Fig.A13b). We observe that DINO has the tendency to overfit to color. On the other hand, CLIP struggles far less with distractors due to its stronger semantic understanding (Fig.A13c).

## A.7   Language-Guided Manipulation

For the language-guided experiments, we distilled CLIP ViT features from `ViT-L/14@336px`. We reuse the 10 demonstrations from the learning to grasp from demonstrations section and their associated $N_q = 100$ query points, and sample $N_r = 8$ rotations for each voxel when initializing the grasp proposals. We minimize the language-guided cost function in Equation 4 for 200 steps with Adam using a learning rate of $2 \times 10^{-3}$.

**Retrieving Demonstrations via Text**   In practice, we compare the user's embedded text query $\mathbf{q}$ with the task embedding $\mathbf{Z}_M$ for each task $M$ which is specified by two demonstrations. We randomly sample object names as our negatives $(L^-)$.

**Inference Time.**   The inference time to optimize for a grasp pose given a language query is 6.9 seconds on average. We did not make any substantial attempts to speed this up, but note that reducing the number of optimization steps (we use 200 steps but observe convergence usually within 50-100 steps), pruning more aggressively, and improving the implementation will significantly reduce inference time.

## A.8   Additional Related Work

We provide a more comprehensive overview of the related work discussed in Section 5.

**Open-Ended Generalization via Language.**   A number of prior work use natural language for task-specification and long-horizon planning in applications such as tabletop and mobile manipulation [23, 26], navigation [27, 28], and more generally, sequential decision-making in games [59, 60]. A recent line of work seeks to replicate the success of vision and language models in robotics, by jointly pre-training large foundation models on behavior data [29, 30]. One can refer to [25] for a

more comprehensive coverage. The goal of our work is different. We seek to find a way to incorporate geometric information with *any* pre-trained features. F3RM is a model-agnostic approach for lifting imagery and language prior knowledge into 3D. In this regard, we are more closely connected to CLIPort [23], which focuses on top-down grasping, and PerAct [24], which trains a voxel grid conditioned behavior transformer from scratch given waypoints in motion trajectories.

**Semantic Understanding and Dense 2D Visual Descriptors.** Learning visual descriptors for dense correspondence estimation is a fundamental problem in computer vision. Among the earliest works, Choy et al. [31] and Schmidt et al. [32] used dynamic 3D reconstruction from RGB-D videos to provide labels of association between pixels from different video frames. Dense Object Nets tackle this problem in the context of robotics [13], and eliminate the need for dynamic reconstruction using multi-view RGB-D images of static scenes [33]. NeRF-Supervision [14] leverages NeRFs for 3D reconstruction, extending descriptor learning to thin structures and reflective materials that pose challenges for depth sensors. Unlike these prior works, recent work in vision foundation models shows that self-supervised vision transformers offer excellent dense correspondence [4]. When scaled to a large, curated dataset, such models offer phenomenal few-shot performance on various dense prediction tasks [34].

**Geometric Aware Representations for Robotics.** Geometric understanding has been a long-standing problem in computer vision [61] and is an essential and mission-critical part of mapping and navigation [35, 36, 37], grasping [38, 39, 17, 40], and legged locomotion [41]. These work either require direct supervision from 3D data such as Lidar point clouds, or try to learn representations from posed 2D images or videos [14]. More recently, the robot grasping community has experimented with neural scene representations to take advantage of their ability to handle reflective or transparent objects and fine geometry [21, 42, 43]. Our work incorporates pre-trained vision and vision-language foundation models to augment geometry with semantics.

**3D Feature Fields in Vision and Robotics** A number of recent work integrate 2D foundation models with 3D neural fields in contexts other than robotic manipulation. For example, Open-Scene [44] and CLIP-Fields [45] distill 2D features into a neural field by sampling points from existing 3D point clouds or meshes. USA-Net [46] and VLMap [47] build 3D feature maps from RGB-D images and VLMs for navigation. Our approach uses RGB images and integrates geometric modeling via NeRF with feature fusion in a single pipeline. ConceptFusion [48] uses surfels to represent the features, which requires $50 - 100s$ GB per scene. Distilled feature fields offer a significant reduction in space. Each of our scenes requires between $60 - 120$ MBs and could be further compressed using lower-rank approximations [62]. Our work shares many similarities with LERF [49]. However, unlike the image-level features used in LERF, which are derived from a hierarchy of image crops, we extract dense, patch-level features from CLIP. Additionally, we take a step further by exploring the utilization of these feature fields for robotic manipulation.

