# OpenReview forum: "Distilled Feature Fields Enable Few-Shot Language-Guided Manipulation"
_robot-learning.org/CoRL/2023/Conference — CoRL 2023 Oral_

### Official Review · Reviewer_j1dX · 2023-07-16

**Confidence:** 3
**Originality:** Good
**Technical Quality:** Very Good
**Clarity Of Presentation:** Good
**Impact:** 3

**Recommendation:**

Weak Accept: I recommend accepting the paper, but will not argue for my recommendation if the majority of other reviewers have a different opinion.

**Review:**

The paper is clearly in a good quality, with well-motivated and well-designed framework. However, I found some explanations in the method section a bit confusing, and some other improvements could be done for the experiments.

1.	I found no problem understanding the framework design up till L112, then it starts to get a bit confusing. I understand the framework starts with constructing a feature neural field with CLIP features, and adding some engineering tricks such as removing the global token and using InstantNGP to improve the training speed and construct high-resolution local information. But later in Section 3.2, I failed to understand i) why and what are the query points; ii) are we learning per-feature 6DOF pose via a learnable map? iii) what and why we need the encoding of a task, and why average the features over the demos represents the encoding for a task? I thought the features should represent task-agnostic semantic information, so we may do object-level grasping?
2.	As using a similar strategy in Neural Descriptor Fields, do we have a prior knowledge knowing that the object we want to grasp has a similar object in the query demo? If so, I believe we need do a proper ablation on how the performance would be changed based on different number/variations of the query demos.
3.	Generalisation over text prompts. Though the paper presents a good generalisation on object categories. I am wondering how the generalisation is on language prompts? Or simply the language prompts her are simply object category names?
4.	Other design choices. As since the proposed network is mainly based on the pre-trained vision-language features. A much simpler and cleaner framework would be just to learn how the transformation from the learned feature map to grasping point based on a text prompt, without using demonstrations as queries, which I believe is a strong baseline.


**Quality Of The Limitations Section:**

Limitations are addressed clearly

**Questions For Rebuttal:**

1. L149 maximising the cosine similarity?
2. Are these 10 demonstrations carefully chosen throughout the entire experiment? How robust it is for a different set of demonstrations?
3. Better explanations for the method section.


**Robotics Focus:**

Sufficient demonstration on hardware

**Summary Of Paper:**

The paper proposes to obtain high-resolution feature fields via InstantNGP and other engineering tricks to improve training time, and then use such feature field to construct 6-DOF poses with iterative optimisation by retrieving the closest poses presented in the demonstrations. The proposed framework is evaluated through real-world grasping tasks, showing good generalisation in novel objects.

**Summary Of Recommendation:**

Good method, but needs some extra clarification. Good results on robotics and video.

---

### Official Review · Reviewer_JsoY · 2023-07-19

**Confidence:** 4
**Originality:** Very Good
**Technical Quality:** Very Good
**Clarity Of Presentation:** Excellent
**Impact:** 4

**Recommendation:**

Weak Accept: I recommend accepting the paper, but will not argue for my recommendation if the majority of other reviewers have a different opinion.

**Review:**

Strengths:
1. The concept of utilizing DFF as a visual representation within robotics appears logical. It's both 3D and semantic-aware, a feat current representations can achieve separately, but not in conjunction.
2. The task design is intelligent, maximizing the benefits of DFFs. The action space is a 6D transformation; with the proposed action reparameterization, the action is also mapped into the 3D feature space, creating homogeneity between the representations of both observation and action.
3. The experimental results successfully demonstrate the proposed method's efficacy.
4. The exposition is exceptionally clear and easy to follow.

Weaknesses:
1. The task design, while inventive, is also somewhat limited. How could this representation be generalized to accommodate closed-loop manipulations?
2. Have alternatives to action reparameterization been considered? For instance, could the query points be set as the midpoints between the two grippers post-transformation?
3. On line 186, could you provide examples of objects with high feature similarities to the demonstrations? Under what circumstances would CLIP outperform DINO?

**Quality Of The Limitations Section:**

Limitations are addressed clearly

**Questions For Rebuttal:**

Please address the queries highlighted in the weaknesses section.

**Robotics Focus:**

Sufficient demonstration on hardware

**Summary Of Paper:**

This paper employs Distilled Feature Fields (DFF) for robotic manipulation. It utilizes DFF as a robust, generalizable, and semantic-aware scene representation. The authors demonstrate this representation's efficacy in tasks such as few-shot manipulation and language-conditioned manipulation.

**Summary Of Recommendation:**

Although the idea is interesting and the experiments are solid, this paper is more about applying CV techniques to robotic tasks with some modification (to make it faster and more available etc.). So I vote for weak accept.

After rebuttal:
I still think the main method is still built on top of *Neural Feature Fusion Fields*, showing interesting applications to robotics. So I keep my vote to weak accept.

---

### Official Review · Reviewer_7BHn · 2023-07-20

**Confidence:** 4
**Originality:** Very Good
**Technical Quality:** Very Good
**Clarity Of Presentation:** Very Good
**Impact:** 4

**Recommendation:**

Strong Accept: I recommend accepting the paper and will argue for my recommendation even if other reviewers hold a different opinion.

**Review:**

**Strengths:**
- The paper introduces a novel approach to derive dense 3D features using a pretrained vision-language model CLIP with NeRF-like representation.
- The paper presents a novel algorithm for integrating both language features as well as stored visual features toward grasping and placing novel objects.
- The paper is well-written and includes supplementary material with details on the distillation procedure and extraction of dense features from CLIP along with the pseudo-code.
- Real-world experiments and results are shown to substantiate the effectiveness of the proposed method.

**Weakness:**
- It is unclear what the paper defines as novel objects. Are these the objects from the same category shown in the demonstration? Or is the work generalizable to a new category of objects? To what degree can the language labels for the objects vary?
- The paper clearly states that the computation is a limitation. It will be beneficial to see the actual time it takes during the inference time upon the query.

**Quality Of The Limitations Section:**

Limitations are addressed clearly

**Questions For Rebuttal:**

Refer to the questions in the "Weakness" section.

**Robotics Focus:**

Sufficient demonstration on hardware

**Summary Of Paper:**

This paper focuses on enhancing robotic manipulation tasks by bridging the gap between 2D image features and the detailed understanding of 3D geometry required for such tasks. The authors leverage NeRF representations to get a dense 3D scene representation. CLIP features are distilled into the NeRF model, allowing it to render features in 3D. The grasping and manipulation demonstrations are connected to 3D feature distributions by observing the pose of the demonstrated grasp and storing the nearby CLIP features and their relative 3D locations. With features from the vision-language model CLIP, the authors demonstrate the capability to designate novel objects for manipulation via free-text natural language. The results show that the model can generalize to novel object instances effectively.

**Summary Of Recommendation:**

I believe the work would be valuable to the community.

**Post-rebuttal update:** The authors have answered my questions. I appreciate the improvement in the quality of the manuscript. My recommendation to accept this paper remains.

---

### Official Review · Reviewer_azcv · 2023-07-20

**Confidence:** 5
**Originality:** Very Good
**Technical Quality:** Excellent
**Clarity Of Presentation:** Excellent
**Impact:** 4

**Recommendation:**

Strong Accept: I recommend accepting the paper and will argue for my recommendation even if other reviewers hold a different opinion.

**Review:**

This paper is well-structured with clear descriptions of the method, experimental results, and limitations. The reviewer finds the supplementary document and the video helpful.

For example, the findings summarized in Table 1 are very reasonable, as CLIP feature contains robust semantic information and DINO feature contains more geometric information. The differences are well supported by the experiment results on the cluttered scene.

In section 3.2, the local geometry transformation is a very effective design for tasks including robotic grasping and manipulation. It might be good to mention the prior work that shares a similar design [NewRef1-2].

References
- Dex-Net 2.0: Deep Learning to Plan Robust Grasps with Synthetic Point Clouds and Analytic Grasp Metrics, RSS 2017.
- Learning 6-DOF Grasping Interaction via Deep Geometry-aware 3D Representations, ICRA 2018.



**Quality Of The Limitations Section:**

Limitations are addressed clearly

**Questions For Rebuttal:**

Besides Figure 8 in the paper, it would be good to provide additional quantitative results and ablation studies (e.g., interpolation method, architecture changes). The reviewer also recommends adding more qualitative results to the final version.

**Robotics Focus:**

Sufficient demonstration on hardware

**Summary Of Paper:**

This paper studies the topic of VLM feature distillation for robotics grasping. Specifically, it proposed a method to distill high-resolution dense 2D features to 3D grids using NeRF. The distilled 3D feature grids is further used in grasping pose matching, with an emphasis on  generalization to open-world objects. Experiments have been conducted on grasping and language-guided object manipulation in the real-world setting. Results demonstrate the effectiveness of the proposed distillation method and strong generalization of the features in grasp pose matching.

**Summary Of Recommendation:**

The reviewer does not find obvious caveats of the paper, besides the limitations (dynamic scenes and the latency) discussed. It is a great work that involves VLM feature, NeRF, and robotic learning all together. I highly recommend accepting the paper at the conference.

---

### Author Response · Authors · 2023-08-12
**General response to all reviewers**

We are grateful to the reviewers for their constructive feedback, and are excited that they recognize the promise in using feature fields to combine 2D features with 3D geometry for robotic tasks.

### Changes based on reviews

We have attached the revised paper and appendix in the response to each reviewer (we are unable to upload files in this official comment). We provide a summary of the changes based on reviewer feedback (highlighted in magenta in the PDF):

- Added additional citations for work that use local geometric transformations for grasping (Reviewer **azcv**).
- Added quantitative results in Appendix A3.1 to compare different feature field architectures (Reviewer **azcv**).
- Provide clarification on definition of novel objects in Sections 4.1 and 4.2 (Reviewer **7BHn**).
- Added inference timing information for language-guided manipulation to Appendix A.7 (Reviewer **7BHn**).
- Added sentence in limitations addressing generalizable NeRFs and closed-loop manipulation (Reviewer **JsoY**).
- Improved clarity for the robot method in Sections 3.2 and 3.3 (Reviewer **JsoY** and **j1dx**).
- Fixed "maximizing cosine similarity" in retrieving relevant demonstrations in Section 3.3 (reviewer **j1dx**).

### General Improvements

We have also made several general improvements to the paper to make the writing and method clearer, including:

- Improved problem formulation for better clarity (Section 2).
- Overhaul of the illustration for the language-guided manipulation pipeline (Figure 3).
- Depicting the full language similarity heatmaps in the top row of Figure 7 for the example execution of language-guided manipulation.
- More detailed and comprehensive related work (Section 5).

These changes have not been highlighted to prevent an excessive amount of highlighting.

### Miscellaneous

The revised paper is currently 9 pages long; however, we will reduce it to 8 pages for the camera-ready version. We additionally created a [website](https://corl-2023-paper334.github.io/) (https://corl-2023-paper334.github.io/) to showcase additional qualitative results and an overview video.

Finally, the title of the paper has been changed to "Distilled Feature Fields Enable Few-Shot **Language-Guided** Manipulation" to emphasize the use of language through vision-language models.

---

### Decision · Program_Chairs · 2023-08-30

**Decision:**

Accept (Oral)

**Comment:**

The paper introduces a novel method for robot manipulation by distilling image features from a vision-language model to 3D scenes. The method enables 6D grasping and language-guided grasping of unseen objects using few-shot demonstrations.

The reviewers achieve a consensus on accepting the paper with high scores. Therefore, I recommend to accept the paper as an oral presentation.